# Experience in different modes of delivery in twin pregnancy

**Jung Chen**[1⊙], **Hung Shen**[1⊙], **Yi Teng Chen**[1⊙], **Chin-Ho Chen**[2‡], **Kuang-Han Lee**[2‡], **Pao-Ling Torng**[1,2]*

1 Department of Obstetrics and Gynecology, National Taiwan University Hospital, National Taiwan University College of Medicine, Taipei, Taiwan, 2 Department of Obstetrics and Gynecology, Hsin-Chu Br, National Taiwan University Hospital, National Taiwan University College of Medicine, Taipei, Taiwan

⊙ These authors contributed equally to this work.
‡ CHC and KHL also contributed equally to this work.
* pltorng@ntu.edu.tw

**Data Availability Statement:** All relevant data are within the manuscript and supplementary information.

**Funding:** The authors received no specific funding for this work.

## Abstract

### Background/purpose

Vaginal delivery, compared with Cesarean delivery, remains a less chosen mode of delivery for twin pregnancy. We studied the maternal and perinatal outcomes of twin pregnancy with different modes of delivery.

### Methods

A retrospective study with data collected from a regional hospital, including vital twin pregnancies delivered at gestational age of 32 weeks and above. Medical charts were reviewed for prenatal conditions and postpartum outcomes.

### Results

Ninety-eight pairs of twins were included and 44.9% were delivered via vaginal delivery. Women in the vaginal delivery group were significantly younger (32.5 ±4.3 years versus 34.8 ±4.6 years, $p < 0.01$), multiparous (34.1% versus 18.5%) and with more twins in vertex-vertex presentation (70.5% versus 33.3%) compared with women in the Cesarean delivery group. There were no differences between maternal postpartum complications and neonatal outcomes in both groups. The outcomes showed longer inter-twin delivery time interval (5.7 ± 5.6 versus 1.5 ± 0.9 min, $p < 0.01$), less estimated blood loss (198.7 ± 144.1 versus 763.2 ± 332.3 mL, $p < 0.01$), and shorter maternal hospital stay (3.0 ± 0.5 versus 5.7 ± 0.5 days, $p < 0.01$) in the vaginal delivery group. Twenty newborns had Apgar score below seven at birth. Logistic regression analysis revealed that low Apgar score was independently related to younger maternal age, maternal obstetric diseases and fetal non-vertex presentation. Gestational weeks and mode of delivery were not related to low Apgar score.

### Conclusion

With careful case selection, vaginal delivery could be safely performed in twin pregnancies with less estimated blood loss and better recovery than Cesarean delivery.

**Competing interests:** The authors have declared that no competing interests exist.

## Introduction

The birth rate of twins in Taiwan has risen substantially over the past decade, accounting for 3.67% of births in the year of 2017 [1]. The rising birth rate of twin pregnancies largely attributes to the thriving of assisted reproductive technology [2]. Twin pregnancies lead to elevated risks for maternal medical and obstetrical complications, including gestational diabetes, hypertensive disorders, and postpartum hemorrhage [3, 4]. Twin pregnancies also contribute to higher perinatal and infant mortality and morbidities, mostly due to increased risk of prematurity [5, 6].

The optimal mode of delivery for twin pregnancies has been a topic of debate. It has been proposed that planned Cesarean delivery for twins may decrease the risk of adverse perinatal outcomes [7, 8]. Based on the United States birth data collected from the National Center for Health Statistics, the rate of Cesarean delivery climbed all the way up to a peak of 75.3% in 2009, and then stabilized with a slight but significant decrease to 74.8% in 2013 [9, 10]. Barrett et al., however, reported in the international randomized trial, "the Twin Birth Study," that planned vaginal delivery showed comparable neonatal mortality and morbidity to planned Cesarean delivery in twin pregnancies [11].

Following the publication of "the Twin Birth Study", both the American Congress of Obstetricians and Gynecologists (ACOG) and the guideline of National Institute for Health and Care Excellence (NICE) suggested vaginal delivery for women with uncomplicated diamniotic twin pregnancies at 32 weeks or later whose presenting fetus is in the vertex position [12, 13]. Despite guideline recommendation, the Cesarean delivery rate for twin pregnancies remains high.

Our study presents birth data from a regional teaching hospital with high vaginal delivery rate in twin pregnancies. We analyzed patient characteristics and the perinatal and maternal outcomes of twin deliveries. We aim to set criteria for safe twin vaginal delivery.

## Methods and materials

### Twin pregnancy

We reviewed all chart records of patients with viable twin pregnancy at gestational weeks of 32 or above, whom delivered their twin babies in Hsin-Chu Branch of National Taiwan University Hospital, a local teaching hospital, from Nov 2013 to Sep2019. Delivery methods, either vaginal or Cesarean section, were stated in chart records based on suggestion of obstetrician and patients' choice after comprehensive explanation of risks and benefits at the outpatient prenatal clinic and during the onset of labor.

Other essential clinical characteristics, including maternal age, body mass index (BMI, calculated as weight in kilograms divided by the square of height in meters), parity, history of previous pregnancy, maternal obstetric diseases (ex. diabetes, gestational diabetes, hypertensive disorder, preeclampsia), underlying maternal medical conditions (ex. hyperthyroidism, uterine myoma, previous myomectomy), and fetal presentations at the last outpatient prenatal visit and during delivery were obtained from medical records. Twin A was defined as the first newborn delivered and twin B the second newborn. Apgar scores of both newborns were recorded at 1 and 5 minutes after delivery. Time intervals between each newborn delivered were recorded. Estimated blood loss (EBL) in Cesarean section was calculated from the amount of fluid loss, including blood and amniotic fluid, collected from the evacuation bottles and on the gauzes used. In vaginal delivery, the amount of amniotic fluid was not included in EBL since most amniotic fluid could not be collected during membrane rupture at time of delivery. All conversions from vaginal delivery to Cesarean section were recorded. After delivery, length of

maternal hospital stay and incidence of immediate intra-operative and postoperative maternal complications such as postpartum hemorrhage and postoperative fever were recorded as well.

Neonatal outcomes were recorded, including admission to neonatal intensive care unit (NICU), respiratory distress syndrome (RDS), transient tachypnea of newborn (TTN), intra-cranial or intraventricular hemorrhage (ICH or IVH), and neonatal death. The criteria of NICU admission included low birth body weight (<1500 g), respiratory distress that needed continuous positive airway pressure or ventilator, unstable vital signs, shock, and neonatal seizure.

### Statistical analysis

Statistical analysis was performed using Statistical Analysis System (SAS) version 8.0 (SAS Institute, Cary, NC). Continuous variables were reported as mean and standard deviation, while discrete variables were reported as percentages. All comparisons of continuous variables were analyzed using t-test, and all discrete variables were analyzed using chi-squared test or Fisher's exact test when sample size was less than five. Logistic regression analyses were performed to identify factors related to poor Apgar score of 7 or lower at either 1 or 5 minutes after delivery. A two-tailed p-value of $< 0.05$ was considered as statistically significant.

### Statement of ethics

This study was approved by the Institutional Review Board of the National Taiwan University Hsin-Chu Hospital (reference no. 108-119-E). All data were fully anonymized before assessment. The ethics committee waived the requirement for informed consent.

### Results

Fig 1 shows the study flow chart. A total number of 98 twins were included. Onewoman with scheduled Cesarean delivery was rearranged to vaginal delivery because her first twin changed spontaneously from breech presentation to vertex presentation at the onset of labor. Three other women's first twins changed to non-vertex presentation during labor onset and were re-scheduled for Cesarean delivery. In total, 44 (44.9%) twins were delivered via vaginal delivery successfully.

Table 1 shows the basic characteristics of these cases in both groups. Women in the vaginal delivery group were significantly younger and showed relatively higher parity than women in the Cesarean delivery group. There were no differences in maternal BMI, obstetric diseases, medical underlying conditions, or gestational age in both groups. During delivery, women in the vaginal delivery group showed significantly more twins in the vertex-vertex presentation. In fact, all first twins in the vaginal delivery group were in the vertex presentation at the time of delivery. There were no significant differences in the presentation of the second twins. In the vaginal delivery group, there were two cases where the second twin received an internal conversion from transverse presentation to a footling or vertex presentation. In six cases, the second twin changed from vertex to either a breech or footling presentation during delivery. No conversions to Cesarean section occurred in the vaginal delivery group. In the Cesarean delivery group, there were five cases where the second twin received an internal conversion from a transverse presentation to either a footling or a vertex presentation. In one case, the second twin was converted from breech presentation to transverse presentation. In nine cases, the second twins were converted from either vertex to breech or breech to vertex presentation during delivery. There were no placenta abnormalities such as abruptio or velamentus placenta during delivery.

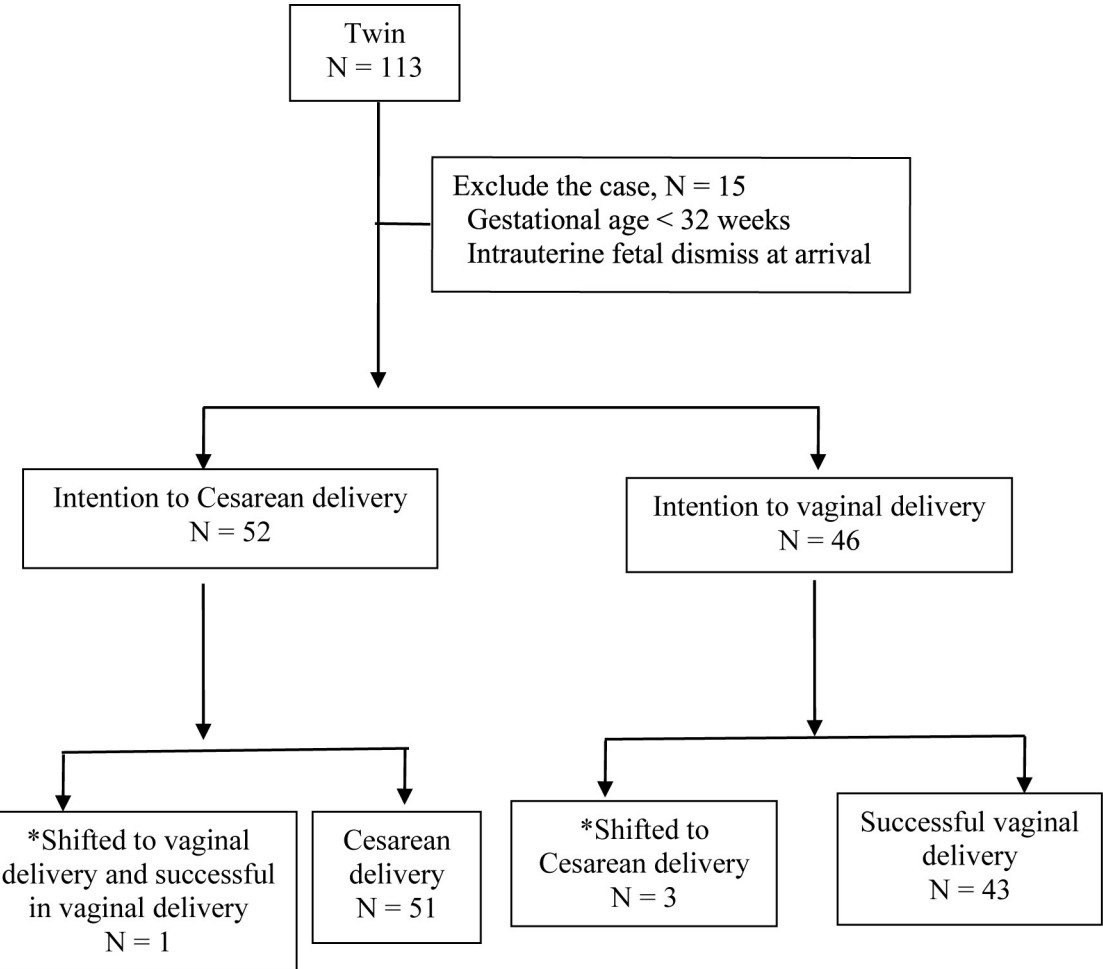

**Fig 1. Flow chart of study population.** * Shift of delivery modes were due to change of first fetal presentation just before delivery.

Table 2 shows the outcome of the mothers and the twin fetuses in both delivery groups. There were no differences in birth weights. There were significantly longer delivery time interval (DTI), less amount of estimated blood loss, and shorter maternal hospital stays in the vaginal delivery group compared with the Cesarean delivery group. Vacuum-assisted delivery was performed in twenty vaginal deliveries. In the four cases where DTI was longer than 10 minutes in the vaginal delivery group, the second twins required an internal conversion to change their presentations from the vertex or transverse presentation to either a footling or breech presentation for fetal extraction. There were no differences in the 1 and 5-minute Apgar scores, set at a cut-off value of 7, in the first and second twins. No neonatal death occurred in either group. TTN was only noted in fetuses of the Cesarean delivery group. There were no significant differences in the rates of other neonatal complications. As for maternal postpartum

**Table 1. Clinical characteristics of the patients.**

|  | Vaginal delivery (n = 44) | Cesarean delivery (n = 54) | *p* value |
|---|---|---|---|
| Maternal Age, years | 32.5±4.3, 20–41 | 34.8±4.6, 26–51 | 0.01* |
| Parity |  |  | 0.13 |
| Nulliparous | 29 (65.9) | 44 (81.5) |  |
| Multiparous | 15 (34.1) | 10 (18.5) |  |
| BMI, kg/m², range | 27.5 ±4.1, 18.5–40.2 | 27.0 ± 6.0, 18.7–37.9 | 0.72 |
| Gestational age, weeks |  |  | 0.50 |
| Preterm (<37wk) | 23 (52.3) | 33 (61.1) |  |
| Term (≥37wks) | 21 (47.7) | 21 (38.9) |  |
| Maternal obstetric diseases |  |  |  |
| Hypertensive disorders | 4 (9.1) | 10 (18.5) | 0.25 |
| GDM or DM | 4 (9.1) | 5 (11.1) | 1.00 |
| Other medical underlying conditions | 1 (2.3) | 3 (5.6) | 0.63 |

Abbreviations: BMI, body mass index; GDM, gestational diabetes mellitus

* $p < 0.05$

Data were shown as mean ± standard deviation, range or n (%)

[a]Two patients in the Cesarean delivery group have hypertensive disorders and GDM; other medical underlying conditions include hyperthyroidism, uterine myoma, previous myomectomy

complications, there was a significantly higher PPH rate in the Cesarean delivery group, but no differences in maternal ICU admission rate or postpartum infection rate.

Twenty newborns reported Apgar scores below 7 at either 1 or 5-minute, or both. Logistic regression analysis revealed that low Apgar score was independently associated with younger maternal age, maternal obstetrical diseases, and fetal non-vertex presentation during delivery. On the contrary, low Apgar score showed no correlation to gestational age or the mode of delivery. The result is shown in Table 3.

## Discussion

Our study presents a high vaginal delivery rate of 44.9% in twin pregnancies in a regional teaching hospital in Taiwan. This rate is exceptionally high compared to data collected from other hospitals in Taiwan, in the Asian countries and even in the world. The vaginal delivery rate in twin pregnancies was reported as 18% in Hong Kong [14], 23–28% in Japan [15], and 25% in the United States [9]. More impressively, our data includes no combined vaginal-Cesarean delivery, no neonatal deaths, and equal complication rates between vaginal and Cesarean deliveries.

A successful and safe delivery for both the mother and twin fetuses is the target goal for clinical service providers. We compared the two groups of patients to determine the characteristics for successful vaginal delivery in twin pregnancies. In concordance with the current ACOG and NICE guidelines, we identified vertex presentation in the first twin as an essential inclusion criterion for patients to receive vaginal twin delivery. In this study, we further identified older maternal age, maternal obstetric diseases, and non-vertex presentation, to be positively related to low Apgar score. Mothers in the vaginal delivery group were significantly younger than the mothers in the Cesarean delivery group. Moreover, multiparity, gestational age, fetal weight, and the mode of delivery were not related to low Apgar score.

However, there was limited data on maternal conditions associated with successful twin vaginal delivery. A prospective cohort study reported that multiparty and spontaneous

**Table 2. Maternal delivery and neonatal outcome.**

| | Vaginal delivery (n = 44) | Cesarean delivery (n = 54) | *p* value |
|---|---|---|---|
| Fetal presentation | | | <0.001*** |
| Vertex/Vertex | 31 (70.5) | 18 (33.3) | |
| Vertex/Non vertex | 13 (29.5) | 12 (22.2) | |
| Non-vertex/vertex | 0 | 11 (20.4) | |
| Non-vertex/non-vertex | 0 | 13 (24.1) | |
| Birth weight, gm | | | |
| Twin A | 2277.7 ± 349.3, 1675–3210 | 2291.5 ± 427.7, 965–3084 | 0.86 |
| Twin B | 2273.0 ± 378.4, 1446–3386 | 2275.6 ± 457.7, 690–3110 | 0.98 |
| Delivery time interval, min | 5.7 ± 5.6, 1–33 | 1.5 ± 0.9, 0–5 | < 0.01** |
| Twin A 1' Apgar Score | | | 0.29 |
| <7 | 2 (4.6) | 6 (11.1) | |
| ≥7 | 42 (95.4) | 48 (88.9) | |
| Twin A 5' Apgar Score | | | 1.00 |
| <7 | 0 | 1(1.9) | |
| ≥7 | 44 (100) | 53 (98.1) | |
| Twin B 1' Apgar Score | | | 0.54 |
| <7 | 6 (13.6) | 5 (9.3) | |
| ≥7 | 38 (86.4) | 49 (90.7) | |
| Twin B 5' Apgar Score | | | 0.45 |
| <7 | 1 (2.3) | 0 | |
| ≥7 | 43 (97.7) | 54 (100) | |
| Estimate blood loss, mL‡ | 198.7 ± 144.1, 100–1000 | 763.2 ± 332.3, 200–1300 | < 0.01** |
| **Maternal complications** | | | |
| 3- or 4-degree perineal laceration | 0 | NA | |
| PPH | 1 (2.27) | 19 (17.59) | <0.0001 |
| ICU admission | 0 | 1 | |
| Infection | 0 | 0 | |
| Maternal Hospital stay, day | 3.0 ± 0.5, 1–4 | 5.7 ± 0.5,5–7 | < 0.01** |
| **Neonatal outcome‡** | | | |
| Admission to NICU | 23 (26.14) | 32 (29.63) | 0.59 |
| ICH/IVH | 1 (1.13) | 3 (2.78) | 0.42 |
| RDS | 6 (6.81) | 9 (8.33) | 0.69 |
| TTN | 5 (5.68) | 18 (16.67) | 0.02 |
| Neonatal death | 0 | 0 | N/A |

Abbreviations: ICU, intensive care unit; ICH, Intracranial hemorrhage; NA, not applicable; NICU, newborn intensive care unit; PPH, postpartum hemorrhage; RDS, respiratory distress syndrome; TTN, transient tachypnea of the newborn

*$p < 0.05$;

** $p < 0.01$;

*** $p < 0.001$

Data were shown as mean ± standard deviation, range or n (%)

‡: total case number of evaluation of neonatal outcome were 88 in vaginal delivery and 108 in Cesarean section

‡ Estimated blood loss in Cesarean delivery group included amniotic fluid

conception best predicted successful twin vaginal birth [16]. Another study reported that advanced maternal age (odds ratio: 2.36) and nulliparity (odds ratio: 5.78) were independently associated with increased likelihood of Cesarean delivery in a univariate analysis [17]. These results correspond with findings in our study.

**Table 3. Logistic regression analyses for determinants of low Apgar Score[a].**

| Independent variables | Univariate Logistic Regression | | | Multivariable Logistic Regression | | |
|---|---|---|---|---|---|---|
| | OR | 95% CI | p-value | OR | 95% CI | p-value |
| Age | 0.915 | (0.822–1.018) | 0.104 | 0.866 | (0.753–0.997) | 0.045* |
| Maternal obstetric disease (PIH, preeclampsia, DM, GDM versus none) | 3.000 | (1.134–7.939) | 0.027* | 4.732 | (1.464–19.292) | 0.009* |
| Parity (multiparous versus primiparous) | 0.970 | (0.334–2.821) | 0.956 | 1.422 | (0.417–4.856) | 0.574 |
| Fetal presentation (Non-vertex versus vertex) | 3.780 | (1.458–9.800) | 0.006* | 3.192 | (1.031–9.881) | 0.044* |
| Gestation age | 0.694 | (0.552–0.871) | 0.002* | 0.766 | (0.525–1.119) | 0.168 |
| Neonatal birth weight | 0.998 | (0.997–1.000) | 0.005* | 1.000 | (1.000–1.001) | 0.634 |
| Types of delivery (Vaginal delivery versus Cesarean delivery) | 0.800 | (0.312–2.053) | 0.643 | 1.012 | (0.304–3.371) | 0.985 |

Abbreviations: CI, confidence interval; DM, diabetes mellitus; GDM, gestational diabetes mellitus; OR, odds ratio; PIH, pregnancy induced hypertension

*: $p < 0.05$;

**: $p < 0.01$

[a]Low Apgar Score was defined as Apgar Score at 1 or 5 min < 7 in either twin.

The presentation of the first twin was an inclusion criterion for vaginal delivery in our study. We found a frequent change of presentation in the second twin during time of delivery in both groups. Houlihan et al. also reported a 20% rate of position change of the second twin in planned deliveries [18]. Successful management of the second twin demands surgical experience. In some conditions, the second fetus needs to be repositioned for a safe and quick delivery. It has been suggested that the high Cesarean delivery rate in twin pregnancies around the world mainly attributes to concern for management of non-vertex presented second twins [19]. Many retrospective studies on vaginal delivery for twin pregnancies were not designed to overcome such situation during patient enrollment. Consequently, planned Cesarean delivery was adopted to reduce the risk of intrapartum anoxia in the second twins [8, 20, 21]. In our study, we have no vaginal Cesarean delivery cases under the technique of internal conversion for safe extraction of the second twin. Luckily, the Twin Birth Study reported no causative relationship between vertex versus non-vertex presentation of the second twin and neonatal mortality and morbidity between the two delivery groups [11].

Twin-to-twin delivery time interval was reported to be negatively correlated to the umbilical cord blood pH, and shorter DTI might improve the neonatal outcome for the second twins [22, 23]. It was reported that DTI over thirty minutes was strongly associated with higher risks of fetal acidosis and low Apgar score in the second twin [24]. However, current guidelines for twin delivery, as stated in ACOG and NICE, offers no specific recommendations regarding optimal DTI [12, 13]. The mean DTI in our vaginal delivery group was 5.7 minutes. The longest DTI was 33 minutes. In the case with the longest DTI, an inexperienced delivery assistant pushed the second twin into a transverse lie after delivery of the first twin. The attending obstetrician performed an internal conversion and delivered the second twin in the breech presentation. The vaginal delivery was successfully without the need for converting into a Cesarean delivery. The Apgar score of the first twin was 6 at 1 minute and 8 at 5 minutes, and 8 at 1 minute and 9 at 5 minutes for the second twin. Both twins were sent to the neonatal intensive care unit for close observation and were later discharged uneventfully. Skillful obstetrical maneuvers such as internal conversions are an essential to prevent combined vaginal-Cesarean delivery and to reduce neonatal morbidity and mortality rates of the second twins. These maneuvers have been reported as standard practice in earlier decades, resulting in a low combined vaginal-Cesarean delivery rate of 0.5% [25]. Unfortunately, many young obstetricians are not familiar or as confident with performing these maneuvers.

As for neonatal outcome, we found more TTN in the Cesarean delivery group. Many previous reports have proved that Cesarean section was a risk factor for TTN. Fluid accumulation in the fetus' lungs after Cesarean delivery due to absence of labor is associated with an increase in TTN [26, 27]. Rates of other severe neonatal complications including RDS and ICH showed no significant difference. These findings are compatible with previous studies that indicated that both modes of delivery were safe for newborns [28, 29]. In our cohort, we found a higher PPH rate in the Cesarean delivery group, which can be attributable to the amount of the blood loss that included amniotic fluid.

The limitation of our study is that it is a non-randomized retrospective study with a small case number. In addition, cases appointed for vaginal delivery were highly selected, such that the first twin must be in the vertex presentation. Despite so, there were no significant differences in fetal weight and gestational week at time of delivery between the two groups. To perform a randomized study with strict criteria of either Cesarean or vaginal delivery in twin pregnancy can be difficult since many unknown events can occur after case recruitment. In our study, we found that a low Apgar score is significantly related to non-vertex presentation (either in the first or the second twin), maternal age, and maternal obstetric diseases. Maternal obstetric diseases can also occur after case randomization. More importantly, as mentioned, fetal presentation can change during delivery. The attending obstetrician must be alert and skillful to handle these situations that can happen in only a few minutes of time. In comparison, in the Twin Birth Study, the rate of Cesarean delivery in the planned vaginal delivery group was as high as 43.8% due to a too early timing of randomization [11, 19].

It should be quite informative if we performed subgroup analyses between the age, parity and presentation matched groups to see possible differences in vaginal delivery and Cesarean delivery. Again, due to limited case number, we failed to obtain significant results on these matters.

Apart from these limitations, we did not have follow-up data of these newborns after delivery. All of the newborns sent to NICU returned home uneventfully. The conditions of these newborns after being discharged were not traced.

The strength of our study is that this is a single regional teaching hospital study with many experienced attending obstetricians as well as a high percentage of twin vaginal delivery with high success rate. To our knowledge, our study provides a useful and complete presentation of a comprehensive data of twin deliveries in Taiwan.

## Conclusions

Our study provides some suggestions for successful vaginal delivery in twin pregnancies. Presentation of first twin is not only the major factor for successful vaginal delivery in twin pregnancies but also an important factor related to Apgar score. With careful case selection and adequate clinical experience, vaginal delivery could be safely performed in twin pregnancies with the additional benefits of less blood loss and better recovery compared with Cesarean delivery.

## Supporting information

**S1 Data.**
(XLSX)

## Acknowledgments

The authors thank Ms. Wan-Chen Chu for her assistance in data acquisition.

## Author Contributions

**Conceptualization:** Chin-Ho Chen, Kuang-Han Lee.

**Data curation:** Hung Shen, Yi Teng Chen, Pao-Ling Torng.

**Formal analysis:** Yi Teng Chen, Pao-Ling Torng.

**Resources:** Chin-Ho Chen.

**Writing – original draft:** Jung Chen, Hung Shen, Pao-Ling Torng.

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
