## [Decision Letter · Decision Letter 0]

9 Oct 2021

PONE-D-21-17824Experience in Different Modes of Delivery in Twin PregnancyPLOS ONE

Dear Dr. Torng,

Thank you for submitting your manuscript to PLOS ONE. After careful consideration, we feel that it has merit but does not fully meet PLOS ONE’s publication criteria as it currently stands. Therefore, we invite you to submit a revised version of the manuscript that addresses the points raised during the review process.

ACADEMIC EDITOR: As noted by some reviewers, “this paper is a very interesting contribution, in a field plagued with a considerable lack of evidence”. But, as noted by Reviewer 3, “it has serious drawbacks that preclude its publication in its current form”. So, to increase the interest for the manuscript, major revisions should be done and the authors should answer to all reviewers’ comments (sample size, potential bias, materials and methods: “the entire section should be re-written coherently with what already happened at the study instead of what they think the editor should read”, said Reviewer 3).

We look forward to receiving your revised manuscript.

Kind regards,

Guillaume Ducarme, MD, MSc, PhD

Academic Editor

PLOS ONE

Journal Requirements:

2. Thank you for including your ethics statement: "The ethical committees of National Taiwan University Hospital, Hsin-Chu Branch (reference no. 108-119-E)"

b) Please provide additional details regarding participant consent. In the ethics statement in the Methods and online submission information, please ensure that you have specified (1) whether consent was informed and (2) what type you obtained (for instance, written or verbal, and if verbal, how it was documented and witnessed). If your study included minors, state whether you obtained consent from parents or guardians. If the need for consent was waived by the ethics committee, please include this information.

4. PLOS requires an ORCID iD for the corresponding author in Editorial Manager on papers submitted after December 6th, 2016. Please ensure that you have an ORCID iD and that it is validated in Editorial Manager. To do this, go to ‘Update my Information’ (in the upper left-hand corner of the main menu), and click on the Fetch/Validate link next to the ORCID field. This will take you to the ORCID site and allow you to create a new iD or authenticate a pre-existing iD in Editorial Manager. Please see the following video for instructions on linking an ORCID iD to your Editorial Manager account: https://www.youtube.com/watch?v=_xcclfuvtxQ.

Reviewers' comments:

Reviewer's Responses to Questions

**Comments to the Author**

1. Is the manuscript technically sound, and do the data support the conclusions?

Reviewer #1: Yes

Reviewer #2: Yes

Reviewer #3: Partly

Reviewer #4: No

2. Has the statistical analysis been performed appropriately and rigorously? 

Reviewer #1: Yes

Reviewer #2: Yes

Reviewer #3: Yes

Reviewer #4: Yes

3. Have the authors made all data underlying the findings in their manuscript fully available?

Reviewer #1: Yes

Reviewer #2: Yes

Reviewer #3: Yes

Reviewer #4: Yes

4. Is the manuscript presented in an intelligible fashion and written in standard English?

Reviewer #1: Yes

Reviewer #2: Yes

Reviewer #3: No

Reviewer #4: Yes

5. Review Comments to the Author

Reviewer #1: The manuscript makes a retrospective study of twin pregnancies in the regional hospital by comparing the vaginal and Cesarean deliveries. This paper is in a good quality to discuss the potential effects of delivery modes on the maternal characteristics as well as the neonatal outcomes.

Major comments:

1. Are there any other placental parameters collected from these Cesarean deliveries?

2. What are the differences between twin A and twin B? Or are they just randomly named? The authors should mention this point.

3. What is the definition of “longer period” in 265?

4. The authors have given an excellent discussion for a retrospective study. However, the advantages of this regional hospital in twin pregnancies can be mentioned by combining with the general local data (ie. historical data from Taiwan or Asia), if applied.

Minor comments:

Please keep a space between the number and the unit. For example, it is suggested to use “<1500 gw” in line 115, and so on.

Reviewer #2: There are differences in age, parity and presentation between groups. This poses a problem in comparing the results between the two groups. Although logistic regression analysis was performed, it would be better to compare subgroup analysis (between the age, parity and presentation matched groups).

Reviewer #3: I must confess: after thoroughly reading this paper a number of times, its potential value chases me. I can foresee this paper as a very interesting contribution, in a field plagued with a considerable lack of evidence. Nonetheless, it has serious drawbacks that preclude its publication in its current form. First of all, the sample size is extremely small for such a large study period (12/13-09/19) in a high reference University Hospital. As being a retrospective cohort, no recruiting data is provided. This, notwithstanding, could be hiding a selection bias. To address this issue I suggest providing further info on the actual amount of twin deliveries at the center during the study period.

The introduction is correct, well written, and coherent. The materials and methods section drove my attention as it suggests they prepared the data collection in a prospective way. Under such a statement this study becomes more of a cross-sectional study, rather than a cohort, even a retrospective one. In my opinion, the entire section should be re-written coherently with what already happened at the study instead of what they think the editor should read.

In the results section, I think the authors devote too much space to explain nonscheduled changes in the fetal presentation at the moment of delivery. Then comes the result that strikes me the most: In the vaginal delivery group TOTAL blood loss was 198 +/- 144 ml, compared to, a far more acceptable value of, 763 +/- 332 ml in the CS group. This is quite an unsustainable finding, dramatically different from those already published by groups studying blood loss during twin births (Whittington JR, et al. AJP Rep. 2020:e330-e334,. Frolova AI, et al Obstet Gynecol. 2016;127:951-956. . Blitz MJ, et al. J Matern Fetal Neonatal Med.;33:3740-3745.). Also, the rate of postpartum hemorrhage in the vaginal delivery group is far too low for being credible unless, again, there is a selection bias underneath.

The discussion is interesting and well sustained. However, it is in a strong need of a style revision by a native English speaker. Also, there is an unnecessary allusion to a particular delivery (lines 226-234) that clearly does not belong there.

The conclusion is far too long and could be summarized in the last paragraph

Reviewer #4: The manuscript entitled "Experience in Different Modes of Delivery in Twin Pregnancy" by Jackie Jung Chen et al. is a retrospective study, which included the data from a regional hospital in Taiwan. The current manuscript compared the difference in the outcome of 98 pairs of twin pregnancy on vaginal delivery and cesarean section. The study revealed that there is not much difference in outcome when compared between vaginal pregnancy and cesarean section. There are multiple grammatical and spelling errors throughout the manuscript.

6. PLOS authors have the option to publish the peer review history of their article (what does this mean?). If published, this will include your full peer review and any attached files.

Reviewer #1: No

Reviewer #2: No

Reviewer #3: **Yes: **Juan Carlos Bello-Munoz

Reviewer #4: **Yes: **Sangappa B Chadchan

---

## [Author Response · Author response to Decision Letter 0]

13 Feb 2022

Thank you for your letter on 29 Jan 2022. I am resubmitting my revised manuscript entitled “Experience in Different Modes of Delivery in Twin Pregnancy.” 

In this revised manuscript, I have made corrections follow reviewer A’s comments. Attached please also find my response to reviewer A’s comment, which is very helpful to improve this manuscript. Please kindly express my sincere gratefulness to the peer-reviewers. 

I am looking forward to hearing from your decision.

---

## [Editor Report · Decision Letter 1]

28 Feb 2022

Experience in Different Modes of Delivery in Twin Pregnancy

PONE-D-21-17824R1

Dear Dr. Torng,

We’re pleased to inform you that your manuscript has been judged scientifically suitable for publication and will be formally accepted for publication once it meets all outstanding technical requirements.

Kind regards,

Guillaume Ducarme, MD, MSc, PhD

Academic Editor

PLOS ONE

Additional Editor Comments (optional): All reviewers comments have been addressed, and that improved the quality of the manuscript.
---

## [Editor Report · Acceptance letter]

3 Mar 2022

PONE-D-21-17824R1 

Experience in Different Modes of Delivery in Twin Pregnancy 

Dear Dr. Torng:

I'm pleased to inform you that your manuscript has been deemed suitable for publication in PLOS ONE. Congratulations! Your manuscript is now with our production department. 

Kind regards, 

on behalf of

Dr. Guillaume Ducarme 

Academic Editor

PLOS ONE